# Hybrid Deep Feature Generation for Appropriate Face Mask Use Detection

**DOI:** 10.3390/ijerph19041939

**Published:** 2022-02-09

**Authors:** Emrah Aydemir, Mehmet Ali Yalcinkaya, Prabal Datta Barua, Mehmet Baygin, Oliver Faust, Sengul Dogan, Subrata Chakraborty, Turker Tuncer, U. Rajendra Acharya

**Affiliations:** 1Department of Management Information, College of Management, Sakarya University, Sakarya 54050, Turkey; emrahaydemir@sakarya.edu.tr; 2Department of Computer Engineering, Engineering Faculty, Kirsehir Ahi Evran University, Kirsehir 40100, Turkey; mehmetyalcinkaya@ahievran.edu.tr; 3School of Management & Enterprise, University of Southern Queensland, Toowoomba, QLD 4350, Australia; Prabal.Barua@usq.edu.au; 4Faculty of Engineering and Information Technology, University of Technology Sydney, Sydney, NSW 2007, Australia; 5Cogninet Brain Team, Cogninet Australia, Sydney, NSW 2010, Australia; 6Department of Computer Engineering, Faculty of Engineering, Ardahan University, Ardahan 75000, Turkey; mehmetbaygin@ardahan.edu.tr; 7Department of Engineering and Mathematics, Sheffield Hallam University, Sheffield S1 1WB, UK; 8Department of Digital Forensics Engineering, College of Technology, Firat University, Elazig 23119, Turkey; sdogan@firat.edu.tr (S.D.); turkertuncer@firat.edu.tr (T.T.); 9School of Science and Technology, Faculty of Science, Agriculture, Business and Law, University of New England, Armidale, NSW 2351, Australia; Subrata.Chakraborty@une.edu.au; 10Centre for Advanced Modelling and Geospatial lnformation Systems (CAMGIS), Faculty of Engineering and Information Technology, University of Technology Sydney, Sydney, NSW 2007, Australia; 11Department of Electronics and Computer Engineering, Ngee Ann Polytechnic, Singapore 599489, Singapore; aru@np.edu.sg; 12Department of Biomedical Engineering, School of Science and Technology, Singapore University of Social Sciences, Singapore 599494, Singapore; 13Department of Biomedical Informatics and Medical Engineering, Asia University, Taichung 41354, Taiwan

**Keywords:** face mask detection, ResNet101, DenseNet201, transfer learning, hybrid feature selector, support vector machine

## Abstract

Mask usage is one of the most important precautions to limit the spread of COVID-19. Therefore, hygiene rules enforce the correct use of face coverings. Automated mask usage classification might be used to improve compliance monitoring. This study deals with the problem of inappropriate mask use. To address that problem, 2075 face mask usage images were collected. The individual images were labeled as either mask, no masked, or improper mask. Based on these labels, the following three cases were created: Case 1: mask versus no mask versus improper mask, Case 2: mask versus no mask + improper mask, and Case 3: mask versus no mask. This data was used to train and test a hybrid deep feature-based masked face classification model. The presented method comprises of three primary stages: (i) pre-trained ResNet101 and DenseNet201 were used as feature generators; each of these generators extracted 1000 features from an image; (ii) the most discriminative features were selected using an improved RelieF selector; and (iii) the chosen features were used to train and test a support vector machine classifier. That resulting model attained 95.95%, 97.49%, and 100.0% classification accuracy rates on Case 1, Case 2, and Case 3, respectively. Having achieved these high accuracy values indicates that the proposed model is fit for a practical trial to detect appropriate face mask use in real time.

## 1. Introduction

Pandemics have occurred throughout human history. The deadliest pandemic was an outbreak of bubonic plague between 1347 and 1352. It caused approximately 30 million deaths, which corresponds to about 40 percent of the population in medieval Europe at that time [1,2]. The first known flu pandemic occurred in the 18th century. During this pandemic, approximately 70% of the world population were infected, but the death rate remained low. Spanish flu was the first pandemic of the 20th century. An outbreak in 1918 caused between 50 to 100 million fatalities. Later, in 1957, Asian flu caused approximately 1 to 3 million fatalities. The first known pandemic of the 21st century was caused by the H1N1 virus. It occurred in 2009 and caused 125,000 to 400,000 deaths [3,4]. On 11 March 2020, an outbreak of COVID-19 was officially classified as pandemic by the World Health Organization (WHO) [5]. The COVID-19 virus was first reported in Wuhan, China, in December 2019 [6]. Since then, the disease has spread rapidly to all communities worldwide because the virus is easily transmitted from person to person by air [7,8]. Crowded environments and a lack of face coverings increases the risk of spreading the virus [9]. The virus has affected 71 million people and caused 1.5 million deaths worldwide as of December 2020 [10].

Face masks are an essential tool to prevent COVID-19 contaminated aerosols and, thereby, slow the virus spread [11]. Hence, wearing face masks in public places has been encouraged by the WHO to control the COVID-19 outbreak [12,13]. Some governments passed laws and regulatory frameworks which introduce the mandatory use of face masks in public places. Even if it is mandatory to wear a mask, some people do not obey this rule for non-permissible reasons [14,15,16]. These people pose a major threat because of their unrestrained ability to spread COVID-19. Therefore, enforcing face covering laws and social standards becomes a priority for governments and local authorities. Prerequisites for law enforcement are adequate detection methods of people who fail to wear face masks. However, face mask detection in public spaces is a hard problem [17]. As such, face masks consist of protective material which is used to cover mouth and nose. This definition allows a wide range of face masks with different visual features. Furthermore, head shape, hair, and the face itself are quite different from person to person. This makes appropriate face mask use detection a hard computer vision problem.

In this study, we propose a machine classification model to automate the detection of appropriate face mask use. We have created a novel Threshold RelieF Iterative RelieF (TRFIRF) algorithm to select features that were extracted with DenseNet201 and ResNet101 from still images. The model was trained and tested with a hand-curated dataset. Each image in the dataset belongs to one of the following three classes: mask (Class 1), no mask (Class 2), and improper mask (Class 3). The model was structured into three main parts: deep feature generation, feature selection using TRFIRF, and classification using support vector machine (SVM). The key contributions of the presented deep hybrid feature- and TRFIRF-based model are given below:1We have curated a new dataset and made it publicly available under: (accessed 20 December 2021) https://websiteyonetimi.ahievran.edu.tr/_Download/MaskDataset.rar.2We have improved the RelieF feature selector by creating an iterative version of that algorithm. Subsequently, we addressed the high time complexity of the iterative RelieF (IRF). The result of these efforts was the TRFIRF algorithm.3A novel transfer learning method for feature generation was created by combining DenseNet201 and ResNet101 with TRFIRF. The extracted features were used to train and test an SVM classifier. The test results indicate that a high-performance face mask detection model was obtained.

The remainder of this paper is organized as follows. The next section provides some background on medical decision support through artificial intelligence. Section 3 details the methods used to design the hybrid deep feature generator for appropriate face mask use detection. After that, Section 4 details the performance results. These results do not stand in isolation. Therefore, in the discussion section, we relate them to findings from other researchers. Furthermore, we introduce limitations and future work before we conclude the paper. 

## 2. Background

Machine learning is a powerful technique used for automatic feature extraction [18,19,20,21]. Many machine learning techniques have been presented in the literature for the detection of different diseases [22,23,24,25,26]. Machine learning techniques, developed especially for the early diagnosis of COVID-19, have achieved successful results [27,28]. Moreover, deep learning models are the most widely used techniques to detect COVID-19, since they are better solutions for COVID-19 classification than feature engineering models [29,30]. Deep learning methods achieved high accuracy when sufficient labeled data was available. Thus, deep learning-based automatic diagnosis systems are of great interest in cases when human expertise is not accessible [31]. Such systems can also serve as adjunct tools to be used by clinicians to confirm their findings. Machine learning methods have been used to detect face masks automatically [32,33]. A wide range of deep learning models, especially convolutional neural networks (CNNs) [34,35], were used to solve computer vision problems. As such, deep learning models are state-of-the-art networks in artificial intelligence, and they are likely to yield high classification performance, even with large datasets. Table 1 presents a selection of recent studies conducted to address the face mask detection problem.

It can be noted, from Table 1, that many databases have been used, and various models were proposed. Most of the developed models delivered high classification performances. To improve and, indeed, to generalize these results, we have incorporated two widely used pre-trained deep learning models, DenseNet201 [58] and ResNet101, [59] to our model as feature generators. A novel TRFIRF algorithm was used as feature selector. These features were fed to an SVM [60,61] classifier.

## 3. Methods

As part of this study, we have designed and implemented a deep feature engineering model to detect facemask-wearing. The main objective of this model was to achieve high classification performance with low time complexity. Therefore, transfer learning has been incorporated as an integral part of the proposed model. Figure 1 provides a schematic illustration of the proposed hybrid deep features and TRFIRF-based face mask detection model. The remainder of this section introduces the individual processing steps in more detail.

In this study, photographs of individuals mask (Case 1), no mask (Case 2), and improper mask (Case 3) were collected by researchers via internet search. The discovered photos were combined with 4072 photos that were uploaded to the Kaggle website by Larxel [62]. A face detection application was created to obtain face images from all the photos in the database. There may be more than one photo of the same individual in the database. This program, coded in C# language, detected automatically faces from photos. Through visual inspection, we eliminated a few low-quality face-mask images. Finally, we collected 529 improper mask, 992 mask, and 554 no mask face images. To explain the used classes, Figure 2 shows example images for each of the three classes.

The attributes of the collected face mask dataset are listed in Table 2.

The collected dataset can be downloaded from (accessed 20 December 2021) https://websiteyonetimi.ahievran.edu.tr/_Download/MaskDataset.rar URL.

As can be seen in Table 2, the used dataset contains 2075 facial images taken from different profiles. By using this dataset, a model for face mask-wearing sensitive doors has been proposed. Moreover, this dataset is a hybrid dataset. We created this facial image dataset using open-source face mask datasets. The most important attribute that distinguishes this dataset from other datasets is the creation of the improper mask class.

Feature concentration was accomplished with two pre-trained deep learning models. ResNet101 [59] and DenseNet201 [58], with 101 and 201 layers, respectively, were used for feature generation [58,59]. Initially, these models were trained using the ImageNet dataset [34], and, by now, these models have been used extensively for transfer learning applications [63,64]. As such, transfer learning models can be used for both feature generation and classification. For our study, we have used this technique for feature generation. To be specific, the fully connected layers of the two pre-trained models were used for this task. Figure 3 shows a block diagram of ResNet101 and DenseNet201. Within that diagram, we have highlighted the layers used for deep feature generation. The following sections provide more details on ResNet101 and DenseNet201.

Numerous studies have shown that CNNs provide good solutions for computer vision problems [65,66]. The CNN network structure is inspired by pyramid cells from the cerebral cortex [67,68]. A drawback of that approach is that CNNs tend to suffer from exploding/vanishing gradient problems; hence, they are difficult to optimize [69]. To solve these problems, various models have been proposed, and one of these models is ResNet. This model architecture uses residual connections which allow some information to bypass specific network layers [59]. The most widely used ResNet implementations are: ResNet18, ResNet31, ResNet50, ResNet101, and ResNet172. These different versions are named after the number of layers. For example, ResNet18 and ResNet31 have 18 and 31 layers, respectively. As such, they are categorized as small networks. In this work, we have used ResNet101, which has 101 layers, for feature generation through transfer learning [59].

Huang et al. [58] presented a densely connected CNN, widely known as DenseNet. It uses hierarchical connections and contains shorter connections. DenseNet uses ResNet, dense connectivity layers, composite function (batch normalization and ReLu), pooling, setting growth rate, bottleneck layers, and compression layers. DenseNet201 can be used for both classification and feature generation. In this work, it was used as a feature generator [58].

In this section, we present our transfer learning approach to feature extraction and selection. Both ResNet101 and DenseNet201 algorithms were used for automated hybrid deep feature extraction. Feature selection was established through the novel TRFIRF algorithm. Finally, SVM was used as classification algorithm. The steps of the proposed model are given below.

Step 0: Load face images.

Step 1: Generate 1000 features by deploying the pre-trained ResNet101.

Step 2: Generate 1000 features by deploying the pre-trained DenseNet201.

Step 3: Merge features and assemble a 2000-dimensional feature vector for each face image.

Step 4: Use the TRFIRF algorithm on the feature vector.

Step 4a: Apply RelieF to the feature vector and obtain weights.

Step 4b: Select features with weights greater than a threshold.

Step 4c: Deploy an iterative RelieF (IRF) to the selected features.

Step 5: Feed the chosen features to the SVM classifier.

In this section, we describe how the deep features were generated from the face images. DenseNet201 and ResNet101 were used in transfer learning mode. The face images were fed to these networks, and 1000 features were obtained from each network. To be specific, these features were obtained from the last fully connected layer (FC1000) of the networks. The primary objective of this phase is to combine the classification abilities of both DenseNet201 and ResNet201, such that this combination outperforms each of the individual classifiers. To conduct this objective, a deep feature engineering model has been introduced, and the processing steps of the deep feature generation algorithm are given below:

Step 0: Load the collected face images.

Step 1: Generate features using pre-trained ResNet101.
(1)$$feat^{ResNet101}=ResNet101\left(Im\right),$$
(2)$$feat^{DenseNet201}=DenseNet201\left(Im\right),$$
where ResNet101(.) and DenseNet201(.) are defined as deep feature generation functions, Im is the image, and $feat^{ResNet101}$ and $feat^{DenseNet201}$ are 1000 dimensional feature vectors.

Step 2: Merge the generated deep features.
(3)$$feat\left(i\right)=feat^{ResNet101}\left(i\right),\;i\in \left\{1,2,\ldots ,1000\right\},$$
(4)$$feat\left(i+1000\right)=feat^{DenseNet201}\left(i\right),$$
where feat defines the concatenated features with a length of 2000.

One of the most important steps during machine learning algorithm design is feature selection. As such, feature selection must establish the feature significance and rank the features accordingly. In this work, we use the novel TRFIRF method, which is a variation of the RelieF [70]. The algorithm creates a feature weighting matrix based on Manhattan distance calculations. The individual weights can be negative and positive. Negative weights represent redundant features. Figure 4 provides a graphical representation of TRFIRF.

The TRFIRF algorithm is composed from two layers. In the first layer, the algorithm calculates the feature weights for threshold-based features selection. In the second layer, the indices of the most relevant features are generated for iterative selection. The steps below indicate how the algorithm functionality unfolds:

Step 0: Apply RelieF to generate 2000 features.
(5)$$w=ReliefF\left(feat,\;act_{out}\right).$$

The ReliefF(.,.) function helps us to explain RelieF algorithm. w defines weights with a length of 2000, and $act_{out}$ represents actual outputs.

Step 1: Select features using the calculated weights (w) and threshold value (trs). In this work, trs was selected as 10^−2^.
(6)$$feat^{T}\left(i\right)=feat\left(count\right),\;if\;w\left(i\right)>trs,\;count=count+1,$$
where $feat^{T}$ represents the selected features using threshold value.

Step 2: Calculate weights of the $feat^{T}$ by using Equation (5).

Step 3: Determine initial and end values. They were selected in between 100 and 500.

Step 4: Select a loss generator. In this work, SVM classifier was used as loss value generator with 10-fold cross-validation.

Step 5: Choose features iteratively.
(7)$$w^{T}=ReliefF\left(feat^{T},act_{out}\right),$$
where $w^{T}$ defines weights of the $feat^{T}$.
(8)$$idx=sort\left(w^{T}\right).$$

In Equation (8), the indices (idx) of the most relevant features are calculated by employing sorting.
(9)$$fv^{k}\left(j\right)=feat^{T}\left(idx\left(j\right)\right),\;j\in \left\{1,2\ldots ,100+k\right\},\;k\in \left\{1,2,\ldots ,400\right\},$$
where $fv^{k}$ the selected kth feature vector.

Step 6: Calculate loss values.
(10)$$loss\left(k\right)=SVM\left(fv^{k},act_{out}\right).$$

Step 7: Calculate the index of minimum loss value.
(11)ind=min(loss).

Step 8: Select the optimal/final features.
(12)$$final=feat^{T}\left(idx\left(j\right)\right),\;j\in \left\{1,2\ldots ,100+idd\right\}.$$

The nine steps, outlined above, define the TRFIRF feature selector.

The final step of the presented face mask detection model is classification. The selected features are fed to the SVM classifier [60,61]. The classifier was trained and tested using 10-fold cross-validation. The features/attributes of the developed SVM classifier are given in the list below:Kernel function: 3rd degree polynomial kernel, also known as Cubic SVM.Kernel scale: Automatic.Box constraint level: One.Coding: One-vs-One.

## 4. Results

The presented model was trained using the dataset described in Section 3. MATLAB (2020a) was used as the programming environment. The model was evaluated based on three test cases. These cases are defined in the text below. In addition, a descriptive view of these cases is presented in Figure 5.

Case 1: Creates a three-class classification problem by using the categories mask, no mask, and improper mask as individual classes. This case contains 2075 images.Case 2: Creates a two-class classification problem by combining wrong mask and no mask to form a ‘non-compliance’ set. Two thousand and seventy-five images were used in this case.Case 3: Creates a two-class classification problem by excluding the improper mask set. This allowed us to compare our results with outcomes from other studies. There are 1546 images in this case.

We have evaluated the classification performance of the SVM model with 10-fold cross-validation. The individual performance parameters were accuracy (ACC), average precision (AP), unweighted average recall (UAR), Mathew correlation coefficient (MCC), F1-score, Cohen’s kappa (CK), and geometric mean (GM) [71,72]. The results obtained for the defined cases are presented in Table 3.

Figure 6 communicates the classification results in the form of a confusion matrix for each of the three cases.

Table 3 shows that the presented model has obtained 100.0% classification accuracy for Case 3, which resulted from 100% accuracy in each of the ten folds. Figure 7 shows the graph of accuracy (%) versus each fold of ten-fold cross-validation for Case 1 and Case 2.

In Figure 7, fold-wise classification accuracies of Case 3 were not depicted since our model has attained 100% classification accuracy on that case. Thus, fold-wise accuracies of Case 3 are equal, and they are 100%.

## 5. Discussion

Compulsory face covering, introduced to slow the spread of COVID-19, significantly impacted on the life of ordinary people worldwide. To reduce the transmission rate, it has become mandatory to wear face masks in some public spaces. However, enforcing that demand is difficult. Systems that can detect people without or with incorrectly worn face masks might help to enforce regulations and, thereby, control the spread of COVID-19. In this work, we propose a tool to address that problem. We use a transfer learning-based feature generation technique to detect face covering violations. To be specific, our method takes still images from faces as input and determines if the person, shown in that image, wears a mask correctly. That functionality was achieved with transfer learning. As part of this study, we developed a feed-forward feature generation model which has a low computational complexity. We have generated the features with two transfer learning algorithms (ResNet101 and DenseNet201). In other words, we have fused two deep learning models. Hence, the resulting feature extractor captured subtle variations in the data which led to a good classification performance. In this paper, various deep networks were tested before ResNet and DenseNet were selected. Pre-trained fully connected network layers were used to obtain features to speed up the model generation. The accuracy (%) obtained using various transfer learning models with our hand-curated face mask image dataset is shown in Table 4.

Table 4 indicates that the best performing transfer learning methods were ResNet101 and DenseNet201. Therefore, we have selected these two CNNs as feature generators. The TRFIRF algorithm was created to facilitate feature selection. Three cases were used to obtain the results. The TRFIRF selected 406, 478, and 345 features for Case 1, Case 2, and Case 3, respectively. Figure 8 shows a graph of loss value versus number of features using the TRFIRF algorithm for Case 1, Case 2, and Case 3.

Figure 8 denotes the iterative feature selection process by stating loss values over the number of features for the three cases. TRFIRF is a parametric feature selection function, and the number of features ranges from 100 to 500. Figure 8 shows that the minimum loss values (close to 0) are obtained for Case 3. Hence, the proposed model yielded the highest classification accuracy of 100.0% (Accuracy = 1-loss value) for Case 3. We take this high classification accuracy value as a strong indication that Case 3 poses the easiest problem. Clearly, mask and no mask image discrimination is easier when compared to the other two problems posed by Cases 1 and 2. In this research, the biggest problem was to detect face images that show people that apply their mask improperly. Therefore, we added the improper mask class to the dataset.

In order to establish that our model has working face mask detection knowledge, we have validated it with the MaskedFace-Net dataset [8]. As such, the MaskedFace-Net is a widely used open access dataset, and it is one of the largest datasets containing correctly/incorrectly mask face images. It contains a combination of actual photographic images and artificial mask images. Table 5 lists the properties of this dataset.

The proposed method achieved 99.75% accuracy when asked to classify the images from the MaskedFace-Net dataset. The high accuracy that was achieved on this large dataset confirms both the performance and the practical applicability of the proposed model. The practical applicability arises from the fact that none of the MaskedFace-Net images was used for training the model.

Unlike the MaskedFace-Net dataset, the dataset we curated contains only completely real images. In other words, our dataset contains real masked and unmasked images. This property distinguishes our dataset from the MaskedFace-Net dataset.

The advantages of this work are given below:A new face mask dataset consisting of real face mask images was developed.A new problem has been defined in this work, and it is named improper mask. Improper wearing of a face mask has been defined as the wrong masked class. Especially in Turkey, improper wearing of a face mask has widely been seen as a common violation of face covering rules. Hence, this unruly behavior is believed to be a major contributor to the spread of COVID-19. To detect this this rule violation, we have defined ‘wrong masked’ as a category in this work. The classification capability of the proposed hybrid deep feature extractor-based model has been demonstrated by using this signal class.Our literature review indicates that most of the face mask-wearing detection methods have been tested on the categories mask and no mask. Our proposal attained 100% classification (magnificent classification capability) accuracy for this case (Case 3). We solved this problem by deploying a hybrid deep feature engineering model. Moreover, we have used transfer learning. Therefore, our model has also low time complexity.A highly accurate deep feature-based model is presented.The presented model used two pre-trained transfer learning networks for feature generation. Therefore, it extracted more salient features with low execution time.A new version of RelieF selector, named TRFIRF, was developed. It selects an optimal number of features automatically.This model can also be used for the automated classification of abnormal classes from normal classes.The disadvantages of this work can be summarized as follows (9 and 10 below).ResNet101 and DenseNert201 networks are not cognitive and lightweight methods. New generation lightweight and cognitive models could be used.Bigger face mask datasets are required to test the model further.

In the future, real time automated face mask detection can be developed with the following steps: (i) public images, collected with wearable and fixed position cameras, (ii) face recognition, (iii) face region segmentation, (iv) face mask detection with the proposed model, and (v) no mask people are reported.

The presented deep feature engineering-based face mask wearing detection application can be used in medical centers and other locations to detect violations of face covering rules. A camera can be placed on the door, and this camera can take a picture of a person’s face that depicts the front profile. That picture can be processed with the presented hybrid deep model. The processing results will indicate possible violations of face covering rules. Deploying such a system will automate and objectify the detection aspect of face covering rule enforcement. A schematic demonstration of our project is shown in Figure 9.

## 6. Conclusions

Automated detection of appropriate mask use based on face images is a challenging and popular problem in machine learning. In this work, an accurate model was developed using deep feature generation, TRFIRF,—based feature selection and classification techniques. We assembled a face mask image dataset which consists of masked, no masked, and improper mask categories. From this dataset, three cases were created. Our proposed model attained 95.95%, 97.49%, and 100.0% accuracies for Case 1, Case 2, and Case 3, respectively. In the future, we aim to extend this work to create a real time face mask detection system. Such a system might reduce the risk of spreading the viruses by monitoring and subsequently enforcing face covering rules.

## Figures and Tables

**Figure 1 ijerph-19-01939-f001:**
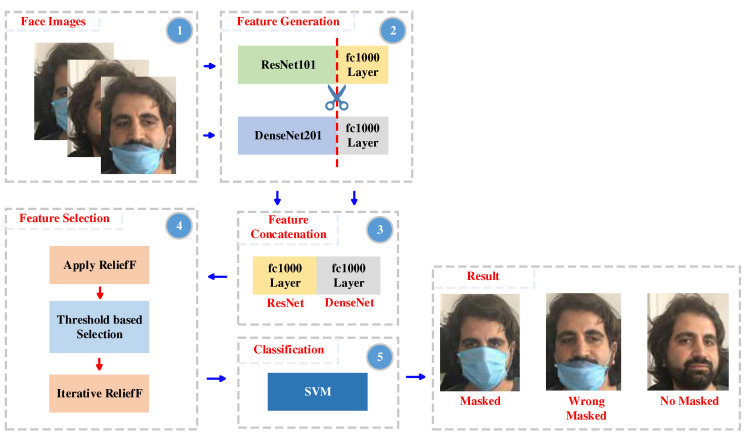
Illustration of the proposed hybrid deep features and TRFIRF-based face mask detection model.

**Figure 2 ijerph-19-01939-f002:**
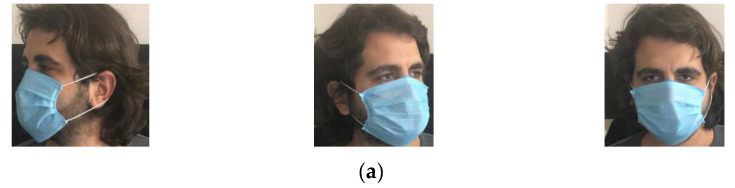
Sample images from the three classes in the dataset: (**a**) mask images, (**b**) no mask images, (**c**) improper mask images.

**Figure 3 ijerph-19-01939-f003:**
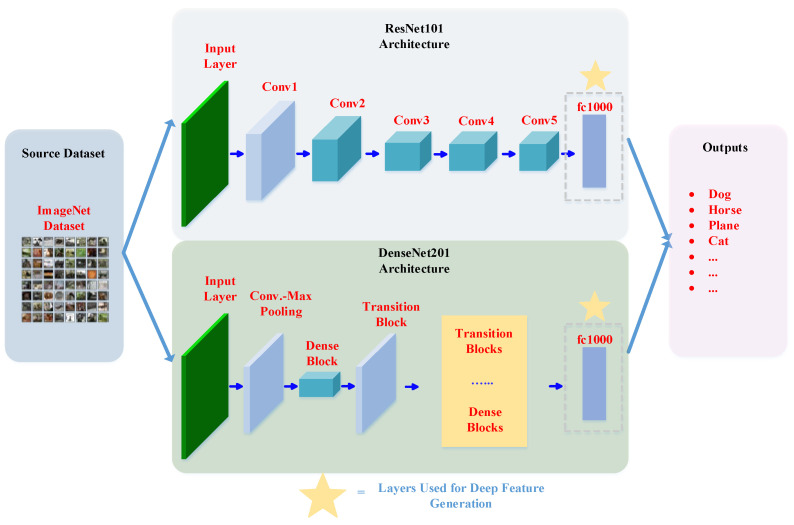
ResNet101 and DenseNet201 deep network architectures.

**Figure 4 ijerph-19-01939-f004:**
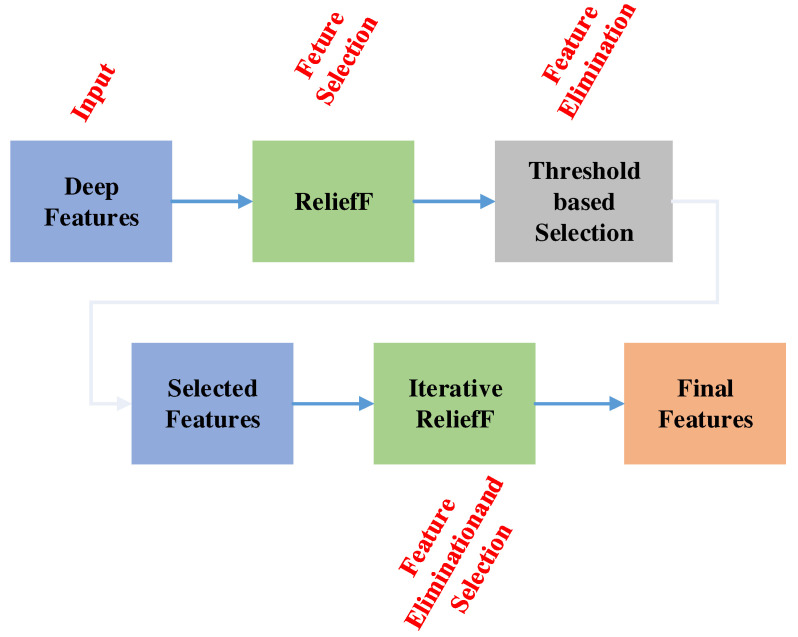
Snapshot of the presented TRFIRF model. In this model, ReliefF is applied two times.

**Figure 5 ijerph-19-01939-f005:**
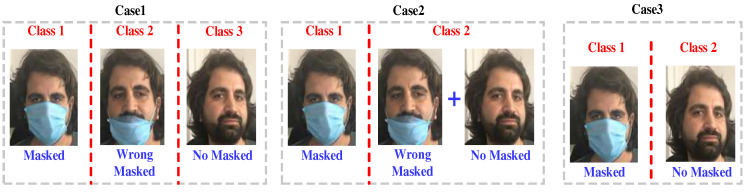
Test cases used in the study.

**Figure 6 ijerph-19-01939-f006:**
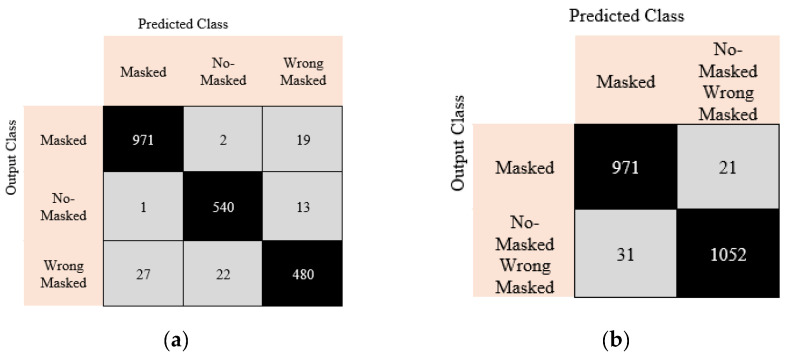
Confusion matrices resulting from training and testing the model with the three different cases: (**a**) Case 1, (**b**) Case 2, and (**c**) Case 3.

**Figure 7 ijerph-19-01939-f007:**
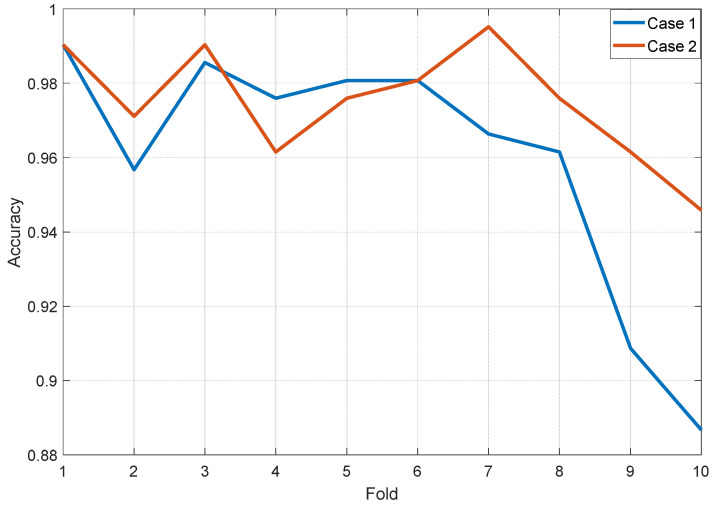
Accuracy (%) versus each fold of ten-fold cross-validation for the Cases 1 and 2.

**Figure 8 ijerph-19-01939-f008:**
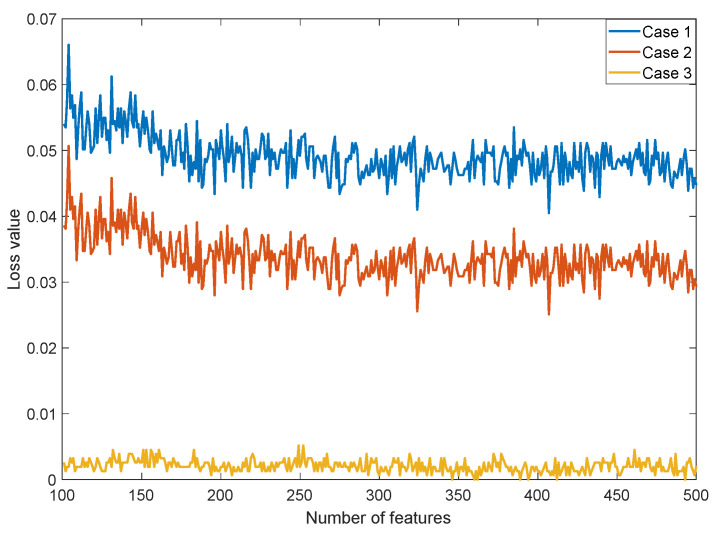
Graph of loss value versus number of features using TRFIRF selector for Case 1, Case 2, and Case 3.

**Figure 9 ijerph-19-01939-f009:**
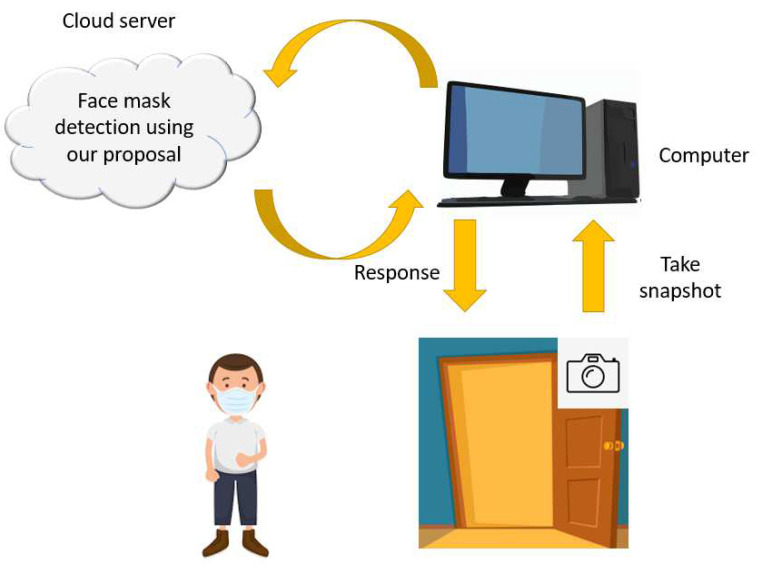
Mask sensitive automatic door.

**Table 1 ijerph-19-01939-t001:** Summary of current studies conducted on face mask detection.

Study	Method	Dataset	Accuracy (%)
Nieto-Rodríguez et al. [36] 2015	Mixture of Gaussians	LFW [37], CMU [38], BAO [39]	95.00
Ejaz et al. [40] 2019	Principal Component Analysis	ORL [41]	72.00
Qin and Li [42] 2020	Super-resolution with classification network	MMD [43]	98.70
Li et al. [44] 2020	You Only Look Once (YOLOv3)	CelebA [45], WIDER FACE [46]	93.90
Hussain et al. [47] 2020	Convolution Neural Networks	KDEF [48]	88.00
Loey et al. [17] 2020	Convolution Neural Networks, Support Vector Machine	RMFD [49], SMFD [50], LFW [37]	100.00
Loey et al. [51] 2020	Convolution Neural Networks, You Only Look Once (YOLOv2)	MMD [43], FMD [52]	81.00
Chowdary et al. [32] 2020	Convolution Neural Networks	SMFD [50]	100
Roy et al. [53] 2020	You Only Look Once (YOLOv3)	Moxa3K [53,54]	63.00
Mohan et al. [55] 2020	Convolution Neural Networks,	FMD [52], FM12kID [56]	99.83
Bhadani and Sinha [57] 2020	Deep Neural Networks, Principal Component Analysis	Collected Data	95.67

**Table 2 ijerph-19-01939-t002:** Amount of class specific data within the dataset.

Classes	Number of Face Images
Mask	992
No masked	554
Improper masked	529
Total	2075

**Table 3 ijerph-19-01939-t003:** Summary of overall performance (%) obtained for the three cases.

Performance Measures	Case 1	Case 2	Case 3
Accuracy (%)	95.95	97.49	100.0
AP (%)	95.56	97.47	100.0
UAR (%)	95.36	97.51	100.0
MCC (%)	93.42	94.98	100.0
F1-score (%)	95.45	97.49	100.0
CK (%)	93.62	94.98	100.0
GM (%)	95.31	97.51	100.0

**Table 4 ijerph-19-01939-t004:** Accuracy results obtained using various transfer learning models with our face mask image dataset. These results were obtained for Case 1.

Number	CNN	Accuracy (%)
1	ResNet101 [59]	93.83
2	DenseNet201 [58]	93.54
3	InceptionResNetv2 [73]	92.72
4	Inceptionv3 [73]	92.43
5	ResNet50 [59]	92.34
6	SqueezeNet [74]	91.90
7	MobileNetv2 [34]	91.04
8	GoogLeNet [75]	90.89
9	ResNet18 [59]	90.70
10	VGG19 [76]	90.51
11	AlexNet [34]	89.93
12	VGG16 [76]	89.88

**Table 5 ijerph-19-01939-t005:** The properties of MaskedFace-Net dataset.

Classes	Number of Face Images
Correctly Masked Face Dataset (CMFD)	67,049
Incorrectly Masked Face Dataset (IMFD)	66,734
Total	133,783

## Data Availability

The data are publicly available: (accessed 20 December 2021) https://websiteyonetimi.ahievran.edu.tr/_Download/MaskDataset.rar.

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
