# Peer review of "Hybrid Deep Feature Generation for Appropriate Face Mask Use Detection"

_ijerph, 2022, doi:10.3390/ijerph19041939_

Round 1
Reviewer 1 Report
Q1: The application scenarios of the proposed algorithm should be introduced. For example, surveillance cameras usually overlook pedestrians from a higher place, while some security systems or web cameras capture human faces from the front. That’s to say, the problems faced by identifying whether a mask is worn correctly or not are different.
Q2: A more comprehensive description of the dataset created should appear. The suggestion in Q1 should also be considered.
Q3: one more question about the created dataset is why did the authors create this dataset? From Section.2, it can be found that there are already more datasets available. What is the difference between them?
Q4: Transfer learning models are used for both feature generation and classification. And what authors want to learn from the ImageNet dataset relevant to your concerns?
Q5: Why not test the algorithm on other datasets to highlight the superiority of your work while making it more convincing?
Q6: Extensive discussion of the problems that remain in relevant existing research is an important way of allowing the reader to learn the significance of your work, rather than simply listing them in section.2.
Q7: The definition of a "Wrong masked" needs to be explained in the manuscript.
Q8: some suggestions about writing in this manuscript are as following:
Line 55:“The COVID-19 virus first appeared in Wuhan, China,…”, ” appeared “should be appropriately changed into ”reported“.
There are too many subheadings in section 3, which are unnecessary from my point.
Line 82-93 contributions of this paper should be re-organized.
Line 75: Acronyms (TRFIRF) should be introduced at their first appearance
Reviewer 2 Report
This paper contains valuable information and is well organized. The authors discussed an important and recent topic. This paper makes a clear scientific contribution about appropriate mask use/detect. The authors should address all of the following minor comments.
- Keywords: We suggest that the authors should replace keywords such as “Deep feature generation” because these keywords are already found in the review article title. It is better that they replace them with other keywords to increase the reach of the article.
- In Table 1, we suggest that the authors add the flaws of the current studies.
- English Writing: This paper requires moderate proofreading to address some typographical and grammatical issues and to make English writing readable and understandable. This revision requires careful proofreading to improve the English writing.
- References list: References are recent and sufficient. However, some search names in the reference list begin an uppercase letter for each word (such as [17], [25], [29] ... etc.) and others use only an uppercase letter in the first word (such as [1], [2], [3] … etc.), authors should standardize style. Some references do not contain enough information such as the references [2], [5], [12] … etc. References should follow the MDPI-IJERPH style. Some terms should be capitalized like “Ieee Access” [28] … etc. Some links do not work in the reference list such as [40] … etc. The reference list needs improvement.
Reviewer 3 Report
This paper proposed a very interesting and useful research for detecting face mask use.
The research design is appropriate and the presentation style is also very clear.
Moreover, the authors created a new dataset of face mask use detection and made it publicly available.
Therefore, I agree to accept this paper for publication.
Reviewer 4 Report
The proposed manuscript investigates the problem of inappropriate use of masks against Covid-19 by a new machine classification model allowing automatic detection of appropriate use of face masks. To study the problem, the authors collected a set of 2075 face mask usage images. The individual images were divided into three groups, namely masked, no-masked, or wrong masked. Based on this division, the authors investigate three cases making comparison between (i) masked vs. no-masked vs. wrong-masked, (ii) masked vs. no-masked + wrong-masked, and (iii) masked vs. no-masked respectively. The authors use the obtained data in their classification model. They describe in detail the primary stages of their method for feature generation (combining two deep learning models), extraction and selection. The proposed method use the chosen features for training and testing a support vector machine classifier. The properties of the proposed model are studied showing very high classification accuracy rates. Having achieved these high accuracy values indicates that the proposed model is fit for a practical trial to detect appropriate face mask use in real time.
The presentation of the results is clear and comprehensive. The authors describe the importance of appropriate face mask use and provide basic information about the medical decision support through artificial intelligence. The methods used to design the hybrid deep feature generator for appropriate face mask use detection are described in detail. The performance results are presented and compared with findings from other studies. The results are valuable and worthy of being published. The obtained results can be used in practice for detection of appropriate face mask use in real time and possible prevention and control of COVID-19 outbreak.
Minor revisions are suggested to improve the quality of the exposition:
p. 1, l. 50-51: I suggest to write “between 50 and 100” instead of “between 50 and 100”;
p. 6, l. 202: The authors should use “,” or “.” in the end of each formula. (Please double check all formulas);
p. 7, l. 228: I suggest to write “Determine initial and end values.” instead of “Determine initial and end value.”
Round 2
Reviewer 1 Report
Authors mentioned in Responding document: “The most important attribute that distinguishes this dataset from other datasets is the creation of the wrong masked class.“, actually, the proposed dataset is not the only one label wrong/incorrect mask. Please refer to the following articles/datasets (not limited to):
1) Wang, J. Zheng and C. L. P. Chen, "A Survey on Masked Facial Detection Methods and Datasets for Fighting Against COVID-19," in IEEE Transactions on Artificial Intelligence, doi: 10.1109/TAI.2021.3139058.
2) MaskedFace-Net – A dataset of correctly/incorrectly masked face images in the context of COVID-19 (nih.gov). https://github.com/cabani/MaskedFace-Net.
3) Batagelj, B.; Peer, P.; Štruc, V.; Dobrišek, S. How to Correctly Detect Face-Masks for COVID-19 from Visual Information? Appl. Sci. 2021, 11, 2070. https://doi.org/10.3390/app11052070
GitHub - borutb-fri/FMLD: A challenging, in the wild dataset for experimentation with face masks with 63,072 face images.
I pretty sure that the paper proposed a very interesting and useful research, but still suggest authors to extend their works to make them more convincing (if possible).
1) tell differences from the similar datasets.
2) to test the algorithm on other datasets to highlight the superiority of your work while.
Author Response
Reviewer 1
Authors mentioned in Responding document: “The most important attribute that distinguishes this dataset from other datasets is the creation of the wrong masked class.“, actually, the proposed dataset is not the only one label wrong/incorrect mask. Please refer to the following articles/datasets (not limited to):
1) Wang, J. Zheng and C. L. P. Chen, "A Survey on Masked Facial Detection Methods and Datasets for Fighting Against COVID-19," in IEEE Transactions on Artificial Intelligence, doi: 10.1109/TAI.2021.3139058.
2) MaskedFace-Net – A dataset of correctly/incorrectly masked face images in the context of COVID-19 (nih.gov). https://github.com/cabani/MaskedFace-Net.
3) Batagelj, B.; Peer, P.; Štruc, V.; Dobrišek, S. How to Correctly Detect Face-Masks for COVID-19 from Visual Information? Appl. Sci. 2021, 11, 2070. https://doi.org/10.3390/app11052070
GitHub - borutb-fri/FMLD: A challenging, in the wild dataset for experimentation with face masks with 63,072 face images.
I pretty sure that the paper proposed a very interesting and useful research, but still suggest authors to extend their works to make them more convincing (if possible).
1) tell differences from the similar datasets.
Response-1: The dataset, which we have collected for the study, consists of real mask images. The datasets in the literature are generally composed of synthetic images. This situation is explained in the discussion section.
Unlike the MaskedFace-Net dataset, we used completely real images. In other words, our dataset contains real mask images. This situation makes our dataset different from the MaskedFace-Net dataset.
2) to test the algorithm on other datasets to highlight the superiority of your work while.
Response-2: The model developed in the study was tested with MaskedFace-Net dataset. The results of this test process have been shared in the discussion section and a copy is given below.
In order to establish that our model has working face mask detection knowledge, we have validated it with the MaskedFace-Net dataset [8]. As such, the MaskedFace-Net is a widely used open access dataset and it is one of the largest datasets containing correctly/incorrectly mask face images. It contains a combination of actual photographic images and artificial mask images. Table 5 lists the properties of this dataset.
Table 5. The properties of MaskedFace-Net dataset.
|
Classes |
Number of face images |
|
Correctly Masked Face Dataset (CMFD) |
67,049 |
|
Incorrectly Masked Face Dataset (IMFD) |
66,734 |
|
Total |
133,783 |
The proposed method achieved 99.75% accuracy when asked to classify the images from the MaskedFace-Net dataset. The high accuracy confirms both the performance and the practical applicability of the proposed model. The practical applicability arises from the fact that none of the MaskedFace-Net images was used for training the model.